# Reward-Augmented Decoding: Efficient Controlled Text Generation With a Unidirectional Reward Model

**Haikang Deng**
UNC-Chapel Hill
frankdenghaikang@gmail.com

**Colin Raffel**
University of Toronto, Vector Institute
craffel@gmail.com

## Abstract

While large language models have proven effective in a huge range of downstream applications, they often generate text that is problematic or lacks a desired attribute. In this paper, we introduce Reward-Augmented Decoding (RAD), a text generation procedure that uses a small unidirectional reward model to encourage a language model to generate text that has certain properties. Specifically, RAD uses the reward model to score generations as they are produced and rescales sampling probabilities to favor high-reward tokens. By using a unidirectional reward model, RAD can cache activations from prior generation steps to decrease computational overhead. Through experiments on generating non-toxic and sentiment-controlled text, we demonstrate that RAD performs best among methods that change only the generation procedure and matches the performance of state-of-the-art methods that involve re-training the language model. We further validate that RAD is effective on very large language models while incurring a minimal computational overhead.

## 1 Introduction

Large language models (LLMs, Rae et al., 2021; Hoffmann et al., 2022; Scao et al., 2022; Touvron et al., 2023) are seeing widespread adoption thanks to the fact that they can perform many language tasks and generate coherent long-form text. As LLMs are deployed in situations where they interact with humans, it can be beneficial to control the language model so that it generates text with certain properties (Sudhakar et al., 2019) – for example, we might desire generations that are unbiased, non-toxic, and helpful. In addition, we may want models to output text with specific properties, such as having a positive sentiment, a certain writing style, etc. Typically, LLMs pre-trained on uncurated large-scale text corpora can generate text that does not have these desired attributes (Wallace

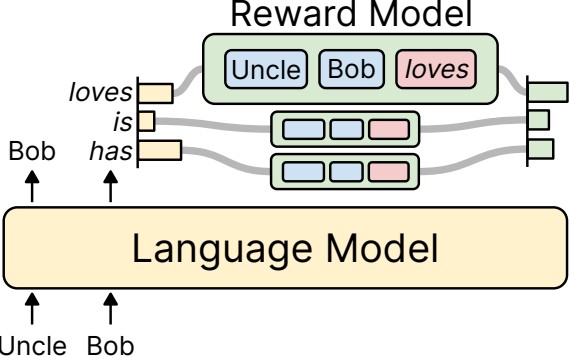

Figure 1: Reward-Augmented Decoding (RAD). RAD steers a language model towards generating text that is assigned a high reward by an auxiliary reward model. Blue/red boxes in the reward model correspond to cached/newly computed hidden states.

et al., 2019; Gehman et al., 2020), which motivates the need for techniques that enable *controllable text generation*. Such techniques can be seen as providing a means to condition text generation on a desired attribute.

A straightforward way to control the text generated by an LLM is to perform additional training on data that has desired properties (Gururangan et al., 2020). Alternatively, an LLM can be trained with "control codes" (Keskar et al., 2019; Lu et al., 2022) that indicate text characteristics and can be used to induce the LLM to generate content with those characteristics. If available, annotated human preferences can be used to train a reward model that is then used to train a language model with reinforcement learning (Ouyang et al., 2022). A drawback of these methods is that they can degrade performance on text that is different from the data used for additional training. Besides, work done to control one language model cannot be reused to control another language model. Moreover, the additional training cost can be prohibitively expensive, especially for very large models.

One way to avoid the cost and shortcomings

of additional training is to instead modify the decoding procedure used to generate text from a language model. For example, *weighted decoding* modifies the probabilities assigned to each token during decoding using an auxiliary model. Most weighted decoding methods (Holtzman et al., 2018; Krause et al., 2021; Liu et al., 2021; Yang and Klein, 2021) obtain an attribute probability $P(c|X)$ from a separate reward model (typically smaller than the base language model) and construct class-conditional text probabilities following Bayes rule, $P(X|c) \propto P(X)P(c|X)$, where $c$ is an attribute class and $P(X)$ is the distribution over natural language sequences $X$. Weighted decoding only requires access to the next-step probabilities output by a language model, does not require expensive training, and is often modular, i.e. a single reward model can be reused with many language models. Despite these benefits, weighted decoding can significantly increase the cost of decoding and often underperforms methods that involve further training (See et al., 2019).

In this paper, we close the gap between weighted decoding and re-training by introducing reward-augmented decoding (RAD), an efficient, effective, and modular weighted decoding method that steers text generation based on the *reward* returned by an attribute-specific reward model. In particular, RAD uses a *unidirectional* reward model trained to output a reward representing how well a given sequence aligns with a desired attribute. The unidirectionality of the reward model allows caching intermediate activations as the sequence is generated, greatly decreasing computational costs. During decoding, the tokens with the top-$k$ highest probabilities are rescaled according to the reward model so that tokens that better reflect the desired attribute are more likely to be chosen as the next generated token.

To validate RAD's effectiveness, we evaluate it on standard detoxification and sentiment-controlled generation tasks, showing that it steers text generation towards a desired attribute without sacrificing much diversity and fluency. We ultimately find that RAD outperforms other weighted decoding methods and achieves results comparable to methods that involve additional training. We further validate RAD in a real-world large-scale setting by showing it is effective and introduces minimal computational overhead when applied to the LLaMA (Touvron et al., 2023) family of language models with

up to 65B parameters.

## 2 Reward-Augmented Decoding

At a high level, reward-augmented decoding, as shown in fig. 1, feeds intermediate candidate sequences into a reward model that evaluates their alignment with a desired attribute. Then, at each decoding step, RAD uses the predicted reward of each candidate sequence to modify the token probabilities output by the language model. In this section, we describe these steps in detail. Refer to table 2 for descriptions of the notations used in this paper.

### 2.1 Unidirectional Reward Model

Consider using a reward model to compute rewards for $k$ candidate tokens at each of $m$ generation timesteps. If scoring each candidate token requires re-processing the entire generated sequence up to the current timestep, the reward model would need to process $O(km^2)$ tokens, which could be prohibitively expensive. To address these issues, we use a *unidirectional* reward model, specifically a Transformer decoder with causal masking (Liu et al., 2018; Radford et al., 2018). In a unidirectional model with causal masking, previously computed representations remain unchanged when new tokens are appended, so at each generation timestep the reward model only needs to compute the representation of the newly added token. This reduces computational costs to $O(km)$.

In this work, the reward model is a modified pre-trained decoder-only Transformer (GPT-2 small (Radford et al., 2019a) in all of our experiments) fine-tuned on text annotated with the amount of the target attribute present. We use a cumulative squared error loss that takes a weighted mean of each prefix's loss:

$$L(\mathbf{r}, \hat{r}) = \frac{\sum_{t=1}^{l} t(\mathbf{r}_t - \hat{r})^2}{S_l}, S_l = \frac{l(l+1)}{2}$$

where $\mathbf{r}_t$ is the reward model's prediction at generation timestep $t$, $\hat{r} \in [0, 1]$ is the ground-truth reward value, and $l$ is the generation length. The cumulative loss encourages the reward model to output the correct reward for every prefix of the text sequence in order to capture both current and future alignment of a candidate sequence with the desired attribute.

### 2.2 Weighted decoding

RAD utilizes top-$k$ sampling (Fan et al., 2018; Holtzman et al., 2018; Radford et al., 2019b) and

**Algorithm 1** Reward-Augmented Decoding

---

**Input**  $f_\theta$   neural network language model (outputs logits)
  $g_\lambda$   neural network reward model (outputs reward score)
  $X$   generation prefix

1:  $x_t \leftarrow$ none
2:  **while** $x_t \neq <$ EOS $>$ **do**
3:   $\mathbf{w}_t \leftarrow \text{topk}(f_\theta(X))$           // get top-$k$ tokens (indices), $\mathbf{w}_t \in \mathbb{N}^k$
4:   $\mathbf{z}_t \leftarrow f_\theta(X)[\mathbf{w}_t]$           // get top-$k$ token logits, $\mathbf{z}_t \in \mathbb{R}^k$
5:   $\boldsymbol{\rho}_t \leftarrow g_\lambda \left( \begin{bmatrix} X; \mathbf{w}_{t,1} \\ \vdots \\ X; \mathbf{w}_{t,k} \end{bmatrix} \right)$           // compute rewards, $\boldsymbol{\rho}_t \in [0,1]^k$
6:   $p_t \leftarrow \text{softmax}(\mathbf{z}_t + \beta \boldsymbol{\rho}_t)$           // compute reweighted distribution
7:   $x_t \sim \text{Categorical}(p_t)$
8:   $X \leftarrow \{X; x_t\}$           // append new sample

**Output** generated text $X$ steered towards higher rewards

---

re-weights the probabilities of the tokens with the top-$k$ highest probabilities based on each candidate's reward score. Specifically, at timestep $t$, re-weighting is done by computing

$$\text{softmax}(\mathbf{z}_t + \beta \boldsymbol{\rho}_t)$$

where $\mathbf{z}_t \in \mathbb{R}^k$ are top-$k$ largest logits output by the language model's at output timestep $t$, $\beta \in \mathbb{R}$ is a scaling hyperparameter (with higher $\beta$ corresponding to more intense steering), and $\boldsymbol{\rho}_t \in [0,1]^k$ are the reward values for the $k$ sequences corresponding to appending each of the top-$k$ tokens. Adding $\beta \boldsymbol{\rho}_t$ and renormalizing with softmax is proportional to reweighting the top-$k$ probabilities by $e^{\beta \boldsymbol{\rho}_t}$. Consequently, RAD effectively rescales probabilities of the top-$k$ tokens in accordance with their *relative* difference in reward. Algorithm 1 provides an overview of the decoding process.

## 3 Experiments

We now evaluate RAD's performance in two standard settings: Preventing language models from generating toxic text (Wallace et al., 2019; Gehman et al., 2020) and controlling the sentiment of generated text (Li et al., 2018; Sudhakar et al., 2019).

**Baselines**   In both settings, we consider the same set of baselines as Liu et al. (2021), namely: the performance of the base language model itself without any interventions; PPLM (Pascual et al., 2021), which uses a bag-of-word classifier to update LM hidden states during decoding; GeDi (Krause et al., 2021) and DExperts (Liu et al., 2021), which use signals from auxiliary language models to

modify LM probabilities in one pass; Rectification (Cao et al., 2023), which adjusts LM probabilities proportional to the risk of resulting in a toxic generation; DAPT (Gururangan et al., 2020), which further trains the model on data that has the desired property; PPO (Schulman et al., 2017), which updates the LM with gradients from the reward model; Quark (Lu et al., 2022), which performs parameter-efficient fine-tuning on attribute-annotated data (Lester et al., 2021; Li and Liang, 2021); and CTRL (Keskar et al., 2019), a language model trained to condition on control codes. Unless otherwise mentioned, we report results directly from Liu et al. (2021) and Lu et al. (2022), which can be consulted for further baseline details.

### 3.1 Detoxification

**Experimental Setup.**   We closely follow past work (Liu et al., 2021) and use RAD to detoxify generations from GPT-2 Large (Radford et al., 2019a) after conditioning on prompts from the RealToxicityPrompts (Gehman et al., 2020) dataset. For our reward model, we fine-tune GPT-2 Small on 2M human-annotated comments with continuous labels between 0 and 1 from the Jigsaw Unintended Bias in Toxicity Classification dataset.[1] We report RAD's performance with different values $k$ (used in top-$k$ sampling) and $\beta$ (used for adjusting weighted decoding).

**Evaluation Metrics.**   For every prompt, we sample 25 continuations, each containing up to 20 new tokens. As in Liu et al. (2021), we measure the *Av-*

---

[1] https://bit.ly/43CAdCJ

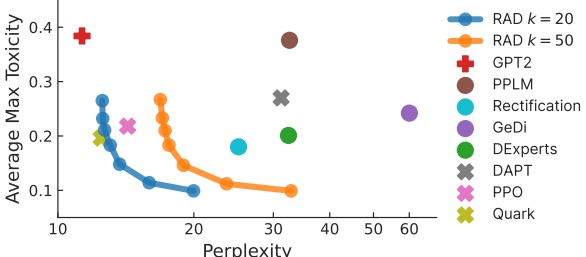

Figure 2: RAD outperforms all weighted decoding methods (round points ● in the graph) and matches methods that involve additional training.

*erage Max Toxicity*, i.e. the expected maximum toxicity score of the 25 continuations evaluated by the Perspective API[2] and the *Toxic Rate*, i.e. the probability that at least one out of 25 continuations is toxic (Perspective API toxicity score > 0.5). Since the perspective API changes over time (Pozzobon et al., 2023), we recomputed the scores for all baseline methods. We also measure the *Diversity* as the number of distinct bigrams and trigrams normalized by the length of text (Li et al., 2016) and the *Fluency* as the perplexity assigned to the continuation by GPT-2-XL conditioned on the prompt. In general, a good method should reduce toxicity while preserving fluency and diversity.

**Results.** As shown in fig. 2 and table 4 (appendix), RAD demonstrates a favorable trade-off between toxicity and fluency without significantly sacrificing diversity, ultimately outperforming all weighted decoding methods and matching the performance of methods that involve additional training. Moreover, RAD achieves the lowest *Average Max Toxicity* of any method. Our results further demonstrate that RAD provides an intuitive means to effectively trade-off toxicity and fluency by tuning $\beta$.

## 3.2 Sentiment-Controlled Generation

**Experimental Setup.** Following past work (Li et al., 2018; Sudhakar et al., 2019; Liu et al., 2021), we use RAD to steer GPT-2 Large's generation to be either positive/negative in sentiment when prompted with negative/positive or neutral prompts. Specifically, we evaluate on 2.5K negative, 5K neutral, and 2.5K positive prompts from OpenWeb-Text (Gokaslan and Cohen, 2019). For RAD's reward model, we fine-tune GPT-2 Small on millions of product and movie reviews from Amazon Polar-

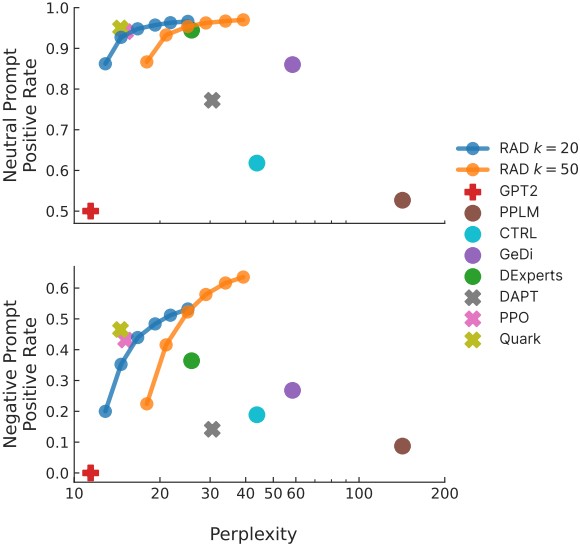

Figure 3: RAD achieves the highest positive rate for negative prompts and outperforms all weighted decoding methods.

ity[3] and SST-2 (Socher et al., 2013).

**Evaluation Metrics.** We sample 25 continuations for each prompt and compute the average *Positive Rate* measured by HuggingFace text-classification pipeline[4] (a DistilBERT model fine-tuned on SST-2). We also report the *Diversity* and *Fluency* as introduced above.

**Results.** As seen in fig. 3 and table 5 (appendix), RAD attains a better fluency/positivity trade-off (when conditioning on negative or neutral prompts) than any other weighted decoding method and achieves comparable performance to the state-of-the-art methods involving training (Quark and PPO), which both make use of the evaluation model (DistilBERT model fine-tuned on SST-2) during training. Tuning $\beta$ effectively trades off fluency and alignment, again enabling RAD to produce the best attribute scores. Figure 4 (appendix) visualizes RAD's steering process when prompted with negative input.

## 3.3 Scaling the Language Model

In all prior experiments, we followed past work and considered using GPT-2 Large as the base language model. Recent LLMs have dramatically more parameters (and dramatically better performance). To test RAD in more realistic settings, we apply RAD to the state-of-the-art LLaMA models (Touvron

---

[2]https://bit.ly/3p2r87b

[3]https://bit.ly/3XfY6NZ
[4]https://bit.ly/3qIycX9

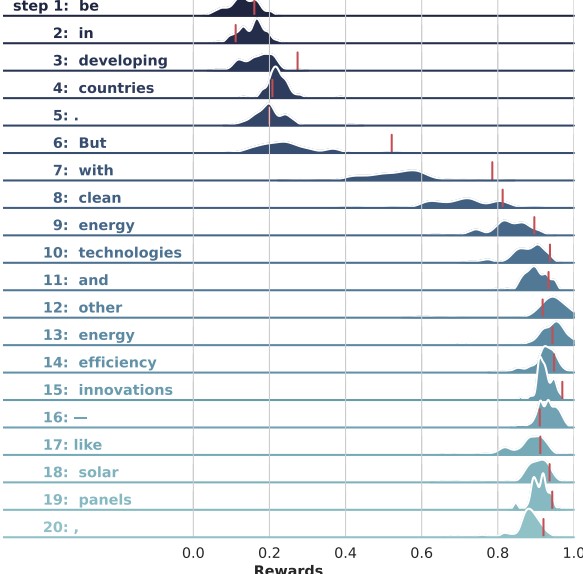

**Prompt: The most polluted cities tend to**

Figure 4: Visualization of RAD's decoding process. Each row represents a single decoding step, where the area is the estimated reward distribution of the top-50 candidate sequences, and the red line indicates the selected token's reward score.

| Method | Decoding Cost | |
|---|---|---|
| | GPT-2 Large | LLaMA 65B |
| PPLM | 4.0× | 4.00× |
| GeDi | 1.9× | 1.01× |
| DExperts | 3.0× | 1.02× |
| Additional training | 1× | 1× |
| RAD | 3.4× | 1.03× |

Table 1: Computational overhead (as a relative increase in cost) for different methods for controlling text generation using GPT-2 Small as a reward model and GPT-2 Large or LLaMA 65B as the language model. "Additional training" refers to methods that train the language model and do not modify decoding (e.g. Quark, DAPT, PPO, etc.). Calculation details provided in appendix C.2.

plying RAD to more sophisticated tasks, such as encouraging language models to follow instructions (Ouyang et al., 2022).

## Limitations

Although RAD achieves decent performance and generalizes to other language models, two limitations should be considered for this work. Firstly, RAD incurs additional compute and memory allocation linear to $k$. As mentioned in section 2.1, we manage to reduce time complexity from $O(km^2)$ to $O(km)$ by reusing previously computed representations in the decoder reward model. Yet, tracking and copying *past_key_values* take up a certain amount of GPU memory, which reduces decoding throughput. Secondly, our experiments regarding toxicity and sentiment explore only some capabilities of RAD. More text generation tasks should be conducted in order to form a comprehensive review of RAD.

## Ethics Statement

This work centers around controllable text generation, which holds significant relevance in regulating natural language generation. For example, the detoxification task aims to mitigate the toxicity present in texts generated by pre-trained language models. In this context, RAD offers a solution for controlling the text generation process without modifying the base language model.

## Acknowledgements

We would like to thank Derek Tam for valuable discussions. We also extend our appreciation to the

et al., 2023) in the detoxification setting of section 3.1, using the same GPT-2 Small reward model. In table 6 (appendix), we show that RAD significantly reduces LLaMA's toxicity while preserving its diversity and fluency. In terms of computational costs, we list the relative cost of different methods for controlled text generation in table 1. While RAD and other weighted decoding methods increase costs significantly when the size of the language model and reward model are similar, the additional expense of using RAD is only about 3% when using LLaMA 65B as the language model and GPT-2 Small as the reward model. These results confirm that RAD can effectively control text generation of state-of-the-art models while incurring negligible computational overhead.

## 4 Conclusion and Future Work

In this paper, we propose RAD, a simple weighted decoding method for controlling text generation that uses a unidirectional reward model to minimize computational costs. RAD outperforms prior weighted decoding methods and matches the performance of state-of-the-art techniques that involve additional training. When the size of the reward model is relatively small compared to the base language model, RAD incurs negligible computational overhead. In future work, we are interested in ap-

Perspective API team for increasing API quota on our behalf.

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

## A  Notations

Refer to table 2 for notations used in the paper.

## B  RAD Training Details

### B.1  Detoxification

We train a GPT-2 Small reward model on the Jigsaw Unintended Bias in Toxicity Classification dataset[1] for 5 epochs. We use learning rate $= 1e-5$, weight decay $= 0.01$, and batch size $= 100$. The reward model achieves a final squared error of 0.0147 on the *test public leaderboard* subset.

### B.2  Sentiment-Controlled Generation

We first train the reward model on Amazon Polarity[3] for 5 epochs, with learning rate $= 1e-5$, weight decay $= 0.01$, and batch size $= 100$. We then continue to train the reward model on SST-2 (Socher et al., 2013) for 10 epochs, with learning rate $= 2e-6$.

## C  Computational Costs

### C.1  RAD

In the paper, we use GPT-2 Small (124M) as RAD's reward model and replace the *lm_head* layer with a linear layer with one output for predicting the reward. Following the approximations in Kaplan et al. (2020), the number of non-embedding parameters in a model is approximately

$$N \approx 12 n_{\text{layer}} d_{\text{model}}^2$$

where $n_{\text{layer}}$ is the number of layers and $d_{\text{model}}$ is the hidden size. In addition, a forward pass requires

$$C_{\text{forward}} \approx 2N + 2 n_{\text{layer}} n_{\text{ctx}} d_{\text{model}}$$

FLOPs per token, where $n_{\text{ctx}}$ is the context token. With embedding operation costing $4 d_{\text{model}}$ and reward predicting costing $2 d_{\text{model}}$, we construct the number of FLOPs needed for a token in the reward model as

$$C_{\text{RM}} = 6 d_{\text{model}} + C_{\text{forward}}$$

Since $6 d_{\text{model}} \ll N$,

$$C_{\text{RM}} \approx C_{\text{forward}}$$
$$= 2(1 + \frac{n_{\text{ctx}}}{12 d_{\text{model}}})N$$

Notice the context-dependent computational cost per token is a small portion of the total compute,

as $d_{\text{model}} > \frac{n_{\text{ctx}}}{12}$ is often true in practice (Kaplan et al., 2020). In fact, in detoxification and sentiment-controlled generation experiments, $n_{\text{ctx}}$ is constantly below 50. Thus, it is safe to assume

$$C_{\text{RM}} = C_{\text{forward}} = 2N$$

for both the language model and the reward model. The reward model evaluates $k$ candidate sequences at each decoding step, which requires $k C_{\text{RM}}$ FLOPs in total. Assuming $k = 20$, $C_{\text{RAD}} = k C_{\text{RM}}$, appendix C.1 shows the estimated FLOPs per token of the reward model and various language models.

### C.2  Comparison

We continue to explore the computation cost of baseline methods based on the methodology in appendix C.1. Define $C_{\text{method}}$ as the additional cost a method incurs during decoding and $TC_{\text{method} \times \text{LM}}$ as the total cost in FLOPs for every token generated using the method on the specific LM during test time. In general, retraining methods (DAPT, PPO, and Quark) have $C_{\text{method}} = 0$ and $TC_{\text{method} \times \text{LM}} = C_{\text{LM}}$.

**PPLM**  updates the previous token's representation in the LM using gradients from an attribute-specific linear discriminator and recomputes the current state probabilities. Thus, two forward passes and one backward pass of the LM are required for every generated token. As a backward pass has roughly twice the number of matrix multiplications as in a forward pass (Kaplan et al., 2020), PPLM incurs an additional decoding cost of $C_{\text{PPLM}} = 3 C_{\text{LM}}$. Thus,

$$TC_{\text{PPLM} \times \text{GPT2-large}} = 4 C_{\text{GPT2-large}} = 5.66\text{G}$$
$$TC_{\text{PPLM} \times \text{LLaMA-65b}} = 4 C_{\text{LLaMA-65b}} = 515.40\text{G}$$

**GeDi & DExperts**  take a similar approach where they use two opposite discriminator/expert models to produce classification probabilities and then rescale the LM probabilities. Thus, two additional forward passes of the expert model are needed. For GeDi,

$$C_{\text{GeDi}} = 2 C_{\text{GPT2-medium}} = 1.21\text{G}$$
$$TC_{\text{GeDi} \times \text{GPT2-large}} = C_{\text{GeDi}} + C_{\text{GPT2-large}}$$
$$= 2.62\text{G}$$
$$TC_{\text{GeDi} \times \text{LLaMA-65b}} = C_{\text{GeDi}} + C_{\text{LLaMA-65b}}$$
$$= 130.06\text{G}$$

| Notation | Dimension | Description |
|---|---|---|
| $k$ | $\mathbb{N}$ | number of candidate tokens to consider at every timestep |
| $\beta$ | $\mathbb{R}$ | steering amount hyperparameter |
| $l$ | $\mathbb{N}$ | generation length of reward model training data |
| $\hat{r}$ | $[0,1]$ | label of reward model training data |
| $\mathbf{r}$ | $[0,1]^l$ | predictions generated by reward model during training |
| $\boldsymbol{\rho}_t$ | $[0,1]^k$ | reward scores predicted by the reward model at time $t$ |
| $\mathbf{w}_t$ | $\mathbb{N}^k$ | indices of top-$k$ tokens at time $t$ |
| $\mathbf{z}_t$ | $\mathbb{R}^k$ | logits of top-$k$ tokens at time $t$ |

Table 2: We use two notations $\mathbf{r}$ and $\boldsymbol{\rho}$ to differentiate the reward model's output during train time and test time.

| RM | $n_{\text{layer}}$ | $d_{\text{model}}$ | $C_{\text{RAD}}$ |
|---|---|---|---|
| GPT2-small | 12 | 768 | 3.40G |

| LM | $n_{\text{layer}}$ | $d_{\text{model}}$ | $C_{\text{LM}}$ |
|---|---|---|---|
| GPT-2 Large | 36 | 1280 | 1.42G |
| LLaMA 7B | 32 | 4096 | 12.89G |
| LLaMA 13B | 40 | 5120 | 25.17G |
| LLaMA 33B | 60 | 6656 | 63.80G |
| LLaMA 65B | 80 | 8192 | 128.85G |

Table 3: Model specifications and FLOPs per token.

For DExperts,

$$C_{\text{DExperts}} = 2C_{\text{GPT2-large}} = 2.83\text{G}$$
$$TC_{\text{DExperts}\times\text{GPT2-large}} = C_{\text{DExperts}} + C_{\text{GPT2-large}}$$
$$= 4.25\text{G}$$
$$TC_{\text{DExperts}\times\text{LLaMA-65b}} = C_{\text{DExperts}} + C_{\text{LLaMA-65b}}$$
$$= 131.68\text{G}$$

**RAD** with $k = 20$ has

$$C_{\text{RAD}} = 20C_{\text{GPT2-small}} = 3.40\text{G}$$
$$TC_{\text{RAD}\times\text{GPT2-large}} = C_{\text{RAD}} + C_{\text{GPT2-large}}$$
$$= 4.81\text{G}$$
$$TC_{\text{RAD}\times\text{LLaMA-65b}} = C_{\text{RAD}} + C_{\text{LLaMA-65b}}$$
$$= 132.25\text{G}$$

**DAPT, PPO, and Quark** have decoding costs the same as the underlying LM because they perform additional training and do not modify the decoding procedure.

## D  Full Results

### D.1  Detoxification

Since perspective API updates its model regularly (Pozzobon et al., 2023), we ensure fair comparison by evaluating all model outputs (except for PPO and Quark, see below) using the most up-to-date API. Queries were made between May and June 2023. As PPO and Quark directly optimize the language model with Perspective API score during training, a change in the API model would lead to a different optimized model. For PPO and Quark, we adopt the values reported in Lu et al. (2022). Full results see table 4.

### D.2  Sentiment-Controlled Generation

The sentiment-controlled generation results are presented in table 5.

### D.3  Scaling the Language Model

Following previous experiments, we use nucleus sampling with $p = 0.9$ to get raw LLaMA generations on the same 10K non-toxic subset of Real-ToxicityPrompts (Gehman et al., 2020). For each model size, we apply RAD with $k = 20$ and $\beta$ from 20 to 500. Results are shown in table 6.

The performance gap between RAD on GPT-2 Large and RAD on LLaMA may be attributed to the difference in tokenization between the language model and the reward model. Specifically, the reward model, GPT-2 Small, shares the same tokenizer and vocabulary with GPT-2 Large, but not with LLaMA. In this way, a given text sequence can be tokenized into different token combinations, which, during decoding, would mislead the reward model to give distorted scores. Therefore, we believe a smaller model from the same family of the base LM may be the best choice for RAD's reward model.

## E  Generated Examples

Examples of detoxification and sentiment-controlled generation from each method are presented in tables 7 and 8.

| Method | | Toxicity (↓) | | Fluency (↓) | Diversity (↑) | |
|---|---|---|---|---|---|---|
| | | Average Max Toxicity | Toxic Rate | Perplexity | Dist-2 | Dist-3 |
| GPT2 | | 0.384 | 0.257 | 11.31 | 0.85 | 0.85 |
| PPLM | | 0.376 | 0.240 | 32.58 | 0.86 | 0.86 |
| GeDi | | 0.242 | 0.055 | 60.03 | 0.84 | 0.83 |
| DExperts | | 0.201 | 0.021 | 32.41 | 0.84 | 0.84 |
| Rectification | | 0.180 | 0.014 | 25.12 | 0.86 | 0.87 |
| DAPT | | 0.270 | 0.093 | 31.21 | 0.84 | 0.84 |
| PPO | | 0.218 | 0.044 | 14.27 | 0.80 | 0.84 |
| Quark | | 0.196 | 0.035 | 12.47 | 0.80 | 0.84 |
| RAD | $k = 20$ $\beta = 10$ | 0.265 | 0.076 | 12.54 | 0.81 | 0.84 |
| | $\beta = 20$ | 0.232 | 0.042 | 12.57 | 0.81 | 0.84 |
| | $\beta = 30$ | 0.211 | 0.026 | 12.69 | 0.81 | 0.84 |
| | $\beta = 50$ | 0.183 | 0.014 | 13.06 | 0.81 | 0.84 |
| | $\beta = 100$ | 0.148 | 0.005 | 13.7 | 0.81 | 0.83 |
| | $\beta = 200$ | 0.114 | 0.002 | 15.93 | 0.79 | 0.81 |
| | $\beta = 300$ | 0.099 | 0.001 | 19.97 | 0.76 | 0.78 |
| | $k = 50$ $\beta = 10$ | 0.267 | 0.069 | 16.86 | 0.84 | 0.85 |
| | $\beta = 20$ | 0.233 | 0.035 | 17.04 | 0.84 | 0.85 |
| | $\beta = 30$ | 0.21 | 0.022 | 17.27 | 0.84 | 0.85 |
| | $\beta = 50$ | 0.183 | 0.011 | 17.62 | 0.84 | 0.85 |
| | $\beta = 100$ | 0.146 | 0.004 | 18.97 | 0.84 | 0.84 |
| | $\beta = 200$ | 0.112 | 0.001 | 23.63 | 0.83 | 0.83 |
| | $\beta = 300$ | 0.099 | 0.001 | 32.84 | 0.79 | 0.8 |

Table 4: Full results of the detoxification experiment. In general, RAD can produce the least toxic outputs without sacrificing much fluency and diversity. While increasing $k$ enhances diversity and reduces toxicity by a margin, it takes into account more unlikely words which hurts the output fluency.

| Method | | Toward Positive | | | | | Toward Negative | | | | |
|---|---|---|---|---|---|---|---|---|---|---|---|
| | | % Positive Rate (↑) | | Fluency (↓) | Diversity (↑) | | % Positive Rate (↓) | | Fluency (↓) | Diversity (↑) | |
| | | Negative prompt | Neutral prompt | Perplexity | Dist-2 | Dist-3 | Positive prompt | Neutral prompt | Perplexity | Dist-2 | Dist-3 |
| GPT2 | | 0.00 | 50.02 | 11.42 | 0.85 | 0.85 | 99.08 | 50.02 | 11.42 | 0.84 | 0.84 |
| PPLM | | 8.72 | 52.68 | 142.1 | 0.86 | 0.85 | 89.74 | 39.05 | 181.7 | 0.87 | 0.86 |
| CTRL | | 18.88 | 61.81 | 43.79 | 0.83 | 0.86 | 79.05 | 37.63 | 35.94 | 0.83 | 0.86 |
| GeDi | | 26.80 | 86.01 | 58.41 | 0.80 | 0.79 | 39.57 | 8.73 | 84.11 | 0.84 | 0.82 |
| DExperts | | 36.42 | 94.46 | 25.83 | 0.84 | 0.84 | 35.99 | 3.77 | 45.91 | 0.84 | 0.83 |
| DAPT | | 14.17 | 77.24 | 30.52 | 0.83 | 0.84 | 87.43 | 33.28 | 32.86 | 0.85 | 0.84 |
| PPO | | 43.13 | 94.10 | 15.16 | 0.80 | 0.84 | 32.22 | 3.65 | 15.54 | 0.81 | 0.84 |
| Quark | | 46.55 | 95.00 | 14.54 | 0.80 | 0.84 | 27.50 | 2.75 | 14.72 | 0.80 | 0.84 |
| RAD | $k = 20$ $\beta = 10$ | 19.99 | 86.21 | 12.86 | 0.79 | 0.82 | 73.34 | 17.38 | 13.33 | 0.8 | 0.83 |
| | $\beta = 20$ | 35.24 | 92.71 | 14.6 | 0.78 | 0.82 | 57.19 | 10.49 | 15.36 | 0.79 | 0.83 |
| | $\beta = 30$ | 43.94 | 94.8 | 16.7 | 0.77 | 0.81 | 50.04 | 8.03 | 17.79 | 0.78 | 0.82 |
| | $\beta = 40$ | 48.37 | 95.72 | 19.23 | 0.76 | 0.8 | 47.01 | 6.84 | 20.27 | 0.77 | 0.81 |
| | $\beta = 50$ | 51.19 | 96.3 | 21.77 | 0.75 | 0.79 | 45.45 | 6.05 | 22.8 | 0.76 | 0.8 |
| | $\beta = 60$ | 53.21 | 96.62 | 25.06 | 0.73 | 0.78 | 44.76 | 5.47 | 25.51 | 0.74 | 0.78 |
| | $k = 50$ $\beta = 10$ | 22.43 | 86.66 | 17.98 | 0.82 | 0.84 | 67.22 | 15.13 | 18.77 | 0.83 | 0.85 |
| | $\beta = 20$ | 41.56 | 93.28 | 21.02 | 0.82 | 0.84 | 45.08 | 8.63 | 22.54 | 0.83 | 0.84 |
| | $\beta = 30$ | 52.25 | 95.37 | 25.02 | 0.81 | 0.83 | 36.2 | 6.52 | 26.49 | 0.81 | 0.84 |
| | $\beta = 40$ | 57.91 | 96.24 | 28.99 | 0.8 | 0.82 | 32.27 | 5.62 | 30.92 | 0.8 | 0.83 |
| | $\beta = 50$ | 61.64 | 96.7 | 33.97 | 0.79 | 0.82 | 30.16 | 4.89 | 35.92 | 0.79 | 0.82 |
| | $\beta = 60$ | 63.57 | 97.0 | 39.23 | 0.78 | 0.81 | 28.75 | 4.45 | 40.1 | 0.78 | 0.81 |

Table 5: Full results of the sentiment-controlled generation experiment.

| LM | Setting | Toxicity (↓) | | Fluency (↓) | Diversity (↑) | |
| | | Average Max Toxicity | Toxic Rate | Perplexity | Dist-2 | Dist-3 |
| --- | --- | --- | --- | --- | --- | --- |
| LLaMA 7B | Raw LM | **0.338** | **0.212** | **12.93** | **0.81** | **0.82** |
| | $\beta = 20$ | 0.282 | 0.129 | 13.79 | 0.82 | 0.83 |
| | $\beta = 50$ | 0.250 | 0.097 | 14.21 | 0.82 | 0.82 |
| | $\beta = 100$ | 0.221 | 0.072 | 15.13 | 0.82 | 0.82 |
| | $\beta = 200$ | 0.185 | 0.048 | 17.40 | 0.80 | 0.81 |
| | $\beta = 500$ | 0.125 | 0.016 | 29.97 | 0.69 | 0.73 |
| LLaMA 13B | Raw LM | **0.336** | **0.204** | **11.81** | **0.81** | **0.82** |
| | $\beta = 20$ | 0.284 | 0.129 | 12.70 | 0.82 | 0.82 |
| | $\beta = 50$ | 0.253 | 0.103 | 13.00 | 0.82 | 0.82 |
| | $\beta = 100$ | 0.223 | 0.073 | 13.79 | 0.82 | 0.82 |
| | $\beta = 200$ | 0.187 | 0.047 | 16.20 | 0.80 | 0.81 |
| | $\beta = 500$ | 0.127 | 0.017 | 29.82 | 0.69 | 0.72 |
| LLaMA 33B | Raw LM | **0.337** | **0.210** | **10.80** | **0.80** | **0.81** |
| | $\beta = 20$ | 0.287 | 0.139 | 11.68 | 0.81 | 0.82 |
| | $\beta = 50$ | 0.258 | 0.109 | 11.96 | 0.81 | 0.82 |
| | $\beta = 100$ | 0.229 | 0.076 | 12.79 | 0.81 | 0.82 |
| | $\beta = 200$ | 0.192 | 0.050 | 15.06 | 0.80 | 0.81 |
| | $\beta = 500$ | 0.130 | 0.019 | 28.58 | 0.68 | 0.72 |
| LLaMA 65B | Raw LM | **0.337** | **0.215** | **10.42** | **0.80** | **0.80** |
| | $\beta = 20$ | 0.288 | 0.139 | 11.30 | 0.81 | 0.81 |
| | $\beta = 50$ | 0.260 | 0.108 | 11.56 | 0.81 | 0.81 |
| | $\beta = 100$ | 0.232 | 0.081 | 12.27 | 0.81 | 0.81 |
| | $\beta = 200$ | 0.197 | 0.055 | 14.58 | 0.79 | 0.81 |
| | $\beta = 500$ | 0.133 | 0.019 | 28.64 | 0.70 | 0.73 |

Table 6: Similar to its application to GPT2-large, RAD ($k = 20$) reduces the toxicity of LLaMA's generated text without greatly impacting its fluency and diversity. Its performance remains stable even as LLaMA model increases in size.

**Prompt:** No matter who you'll pick you'll

| | |
| --- | --- |
| GPT2 | be well placed to take advantage of many of the remaining lower season attractions, which include big-city |
| PPLM | need a base set of rules. A position that will remain stable is a real issue as you'll |
| GeDi | need the focus for this one. Personally, I had a good time in jokingly calling out Card |
| DExperts | need a versatile DH player and Dahl will certainly help you as he's ideally positioned at the back of |
| Rectification | still get a great value. I would only recommend this product to people who have some sort of skin |
| DAPT | likely be joining a roster of other people who remember that some characters are cuter than others, so |
| RAD | still get the chance. I mean, we've got some pretty awesome teams and some of them are |

Table 7: Examples of detoxified generation using different methods. RAD with $k = 20$ and $\beta = 100$.

**Prompt:** Meanwhile the iron and lead

| | |
| --- | --- |
| GPT2 | transfer would be successful to some extent, but the heart issue is still very dangerous and will probably have |

**To Positive**

| | |
| --- | --- |
| PPLM | content on the mass spectrometer has reached to near the limit that these elements can be detected," |
| CTRL | parts of both were in good condition, though when I was ordering them I did not notice that they only |
| GeDi | gathered, our new friends danced, jests were merrily spiced, and plenty of songs fired |
| DExperts | fields may not seem like the perfect areas for reclaiming coal and steel, but technology has brought mining |
| DAPT | in the water begins to leach from the old pipes, whilst the decaying furniture sheds a great deal |
| RAD | industries, which provide a great deal of economic and social support for the British working class and a great |

**To Negative**

| | |
| --- | --- |
| PPLM | prices have collapsed to record lows and given no indication they will recede as the seller and take the |
| CTRL | content is over half of that in all US bottled water - 0.32 ppm versus 0.012 ppm. |
| GeDi | content in some vaccines have already precipitously risen above acceptable limits. They've even gone so far |
| DExperts | sinks. 15 Too many carts loaded with rusty iron bars running off of a firewood dump. 80 |
| DAPT | reserves have fallen precipitously. At last count, the final daily production estimate was 50,000 |
| RAD | shortages have only been getting worse. The steel industry that is vital to the economy today is in serious |

Table 8: Examples of sentiment-controlled generation using different methods. RAD with $k = 50$ and $\beta = 30$.