# OpenReview forum: "Reward-Augmented Decoding: Efficient Controlled Text Generation With a Unidirectional Reward Model"
_EMNLP/2023/Conference — EMNLP 2023 Main_

### Official Review · Reviewer_Ekrw · 2023-08-05

**Soundness:** 4

**Excitement:**

4: Strong: This paper deepens the understanding of some phenomenon or lowers the barriers to an existing research direction.

**Missing References:**

None

**Paper Topic And Main Contributions:**

The method that controls a language model so that it generates text
with certain properties is crucial. This paper presents a method for
Reward-Augmented Decoding (RAD), where a small unidirectional reward
model is used to encourage the language model to generate text that
has certain properties.  By using the unidirectional reward model, the
caching intermediate activations is enable, which leads to an
efficient decoding. The experimental results on standard
detoxification and sentiment-controlled generation tasks demonstrate
that the proposed method outperforms other weighted decoding methods
and achieves results comparable to re-training methods.

Although the proposed method is simple, the proposed method is
superior both in performance and decoding speed from the experimental
results. The paper is relatively easy to follow.


**Questions For The Authors:**

- In the experiments presented in Section 3.3 (Table 5), is k 20 or 50?


**Reasons To Accept:**

- The proposed method is superior both in performance and decoding
  speed in comparison with other weighted decoding methods and
  re-training methods.

- The paper is relatively easy to follow.


**Reasons To Reject:**

- It is not clear why the proposed method achieves better performance
  (e.g., average max toxicity) in comparison with other weighted
  decoding methods and achieves results comparable to re-training
  methods. Discussion is required for this point.


**Reproducibility:**

4: Could mostly reproduce the results, but there may be some variation because of sample variance or minor variations in their interpretation of the protocol or method.

**Reviewer Confidence:**

4: Quite sure. I tried to check the important points carefully. It's unlikely, though conceivable, that I missed something that should affect my ratings.

**Typos Grammar Style And Presentation Improvements:**

- l073: Add the notations for c and X

- Figure 1: Add a mention to Fig. 1 in the main text

- l141: Why $r_t$ is bold?

- l227: fig. 2 -> Fig. 2, table 3 -> Table 4

---

> ### Author Rebuttal · Authors · 2023-08-29
>
> Thank you for reviewing our paper and for your helpful suggestions for improving the quality and readability of our paper. We decided to update the notations and add a pseudocode of RAD to enhance clarity.
>
>
> | Notation | Dimension | Description |
> |:-:|-|-|
> |$\beta$ | $\mathbb{R}^+$ | steering amount hyperparameter |
> |$l$ | $\mathbb{N}$ | generation length of reward model training data |
> |$\hat r$ | $[0,1]$ | label of reward model training data |
> |$\mathbf{r}$ | $[0,1]^l$ | predictions generated by reward model during training |
> |$\mathbf{w}_t$ | $\mathbb{N}^k$ | indices of top-$k$ tokens at time $t$ |
> |$\mathbf{z}_t$ | $\mathbb{R}^k$ | logits of top-$k$ tokens at time $t$ |
> |$\boldsymbol{\rho}_t$ | $[0,1]^k$ | reward scores predicted by the reward model at time $t$ |
>
> **Table 1:** Updated notations. We use two different notations $\mathbf{r}$ and $\boldsymbol{\rho}$ to differentiate the reward model's output in train time and test time.
>
> &nbsp;
>
> ---
> Algorithm 1:  Reward-Augmented Decoding
> ---
> **Input:** $f_\theta$, neural network language model (outputs logits) \
> &nbsp;&nbsp;&nbsp;&nbsp;&nbsp;&nbsp;&nbsp;&nbsp;&nbsp;&nbsp;&nbsp;&nbsp;&nbsp;$g_\lambda$, neural network reward model \
> &nbsp;&nbsp;&nbsp;&nbsp;&nbsp;&nbsp;&nbsp;&nbsp;&nbsp;&nbsp;&nbsp;&nbsp;&nbsp;$X$, generation prefix
>
> 1. $x_t \leftarrow \mathtt{none}$
> 2. **while** $x_t \ne \mathtt{<EOS>}$ **do**
> 3. &nbsp;&nbsp;&nbsp;&nbsp;
>     $\mathbf{w}_ t \leftarrow \texttt{topk}(f_\theta(X))$
>     &nbsp;&nbsp;&nbsp;&nbsp;&nbsp;&nbsp;&nbsp;&nbsp;&nbsp;&nbsp;&nbsp;&nbsp;
>     // get top-$k$ tokens (indices), $\mathbf{w}_t \in \mathbb{N}^k$
> 4. &nbsp;&nbsp;&nbsp;&nbsp;
>     $\mathbf{z}_ t \leftarrow f_\theta(X)[\mathbf{w}_t]$
>    &nbsp;&nbsp;&nbsp;&nbsp;&nbsp;&nbsp;&nbsp;&nbsp;&nbsp;&nbsp;&nbsp;&nbsp;&nbsp;&nbsp;&nbsp;&nbsp;&nbsp;&nbsp;&nbsp;
>     // get top-$k$ token logits, $\mathbf{z}_t \in \mathbb{R}^k$
> 5. &nbsp;&nbsp;&nbsp;&nbsp;
>     $\boldsymbol{\rho}_ t \leftarrow g_\lambda \left(\begin{bmatrix} X;\mathbf{w}_ {t,1} \\\\ \vdots \\\\ X;\mathbf{w}_ {t,k} \end{bmatrix}\right)$
>     &nbsp;&nbsp;&nbsp;&nbsp;
>     // compute rewards, $\boldsymbol{\rho}_t \in [0,1]^k$
> 6. &nbsp;&nbsp;&nbsp;&nbsp;
>     $p_t \leftarrow \mathtt{softmax}(\mathbf{z}_t+\beta \boldsymbol{\rho}_t)$
>     &nbsp;
>     // compute reweighted distribution
> 7. &nbsp;&nbsp;&nbsp;&nbsp;
>     $x_t \sim \mathrm{Categorical}(p_t)$
> 8. &nbsp;&nbsp;&nbsp;&nbsp;
>     $X \leftarrow \\{X;x_t\\}$
>     &nbsp;&nbsp;&nbsp;&nbsp;&nbsp;&nbsp;&nbsp;&nbsp;&nbsp;&nbsp;&nbsp;&nbsp;&nbsp;&nbsp;&nbsp;&nbsp;&nbsp;&nbsp;&nbsp;&nbsp;&nbsp;&nbsp;&nbsp;
>     // append new sample
>
> **Output:** generated text $X$ steered towards higher rewards
>
> ---
>
> &nbsp;
>
> According to your comment, we have made the following changes:
> - On line 143, we clarified that $l$ is supposed to be the **generation length** instead of the current generation timestep.
> - We further clarified that $c$ is an attribute class. $P(X)$ is the distribution over natural language sequences $X$. $P(X|c)$, the distribution over $X$ conditioned on an attribute class $c$, is proportional to $P(X) P(c|X)$.
> - We added a mention of Fig. 1 in the opening paragraph of Section 2.
> - In the equation, $\mathbf{r} \in [0,1]^l$ is a vector and $\mathbf{r}_t$ is a scalar.
>
> As per your question, Table 5 is $k$=20. We added this information to the table description.
>
> In addition, we would like to provide more discussion on (1) why RAD is superior to other weighted decoding methods, and (2) why RAD is comparable to retraining methods.
>
> (1) **Why is RAD superior to other weighted decoding methods?**\
> &nbsp;&nbsp;&nbsp;&nbsp;First, note the different uses of the small auxiliary model in RAD compared to other strong weighted decoding methods such as GeDi and DExperts—we use the auxiliary model as a reward model to discriminate candidate sequences at each intermediate step whereas GeDi and DExperts use the auxiliary model as a class-conditional language model to steer generation. One reason that RAD's approach works better could be that classification is an easier task as compared to class-conditional language modeling, allowing small discriminators to be more helpful than small expert LMs in guiding language generation.\
> &nbsp;&nbsp;&nbsp;&nbsp;Second, RAD’s reward model makes more use of label information if it is not a binary score (i.e. a fraction in [0,1]). RAD’s reward model is trained to minimize the squared error loss while the auxiliary model in GeDi and DExperts does not utilize the label score efficiently—they use label score primarily for data filtering for generative training.\
> &nbsp;&nbsp;&nbsp;&nbsp;Another factor that may have contributed to RAD’s superior performance is the weighted cumulative loss mentioned in section 2.1. During reward model training, by taking every prefix’s reward score into account, we encourage the reward model to converge to the correct reward score early in the sequence. In this way, the reward model is trained not only to capture the current but also the future alignment score of a generation course, which better steers the generation.
>
> (2) **Why is RAD comparable to retraining methods?**\
> &nbsp;&nbsp;&nbsp;&nbsp;While we don’t have a direct answer for this questions due to the essential difference in the design of RAD and retraining-based aligning methods, we would like to point out that, as suggested by Korbak et al., (2022) [1] , Bayesian inference (as used in RAD and most weighted decoding methods), if optimal, would produce an equivalent controlling effect as KL-bounded RL retraining methods (such as reinforcement learning from human feedback). We believe that, in theory, weighted decoding methods are not inferior to retraining methods in aligning language models, and therefore it is plausible that RAD produces results comparable to that of some retraining methods.
>
> [1] Korbak et al., "RL with KL penalties is better viewed as Bayesian inference", in Findings of the Association for Computational Linguistics: EMNLP 2022.

---

### Official Review · Reviewer_GRYE · 2023-08-05

**Soundness:** 3

**Excitement:**

3: Ambivalent: It has merits (e.g., it reports state-of-the-art results, the idea is nice), but there are key weaknesses (e.g., it describes incremental work), and it can significantly benefit from another round of revision. However, I won't object to accepting it if my co-reviewers champion it.

**Paper Topic And Main Contributions:**

In this paper, the authors propose to use the reward models directly in the decoding phase to avoid the additional training cost. Specifically, they follow the weighted decoding method framework, where the logits are adjusted by an additive reward value from an additional reward model. They conduct experiments on detoxification and sentiment-controlled generation tasks. The results show that the proposed method outperforms other reweighting techniques.

The highlights of this paper include:
* The proposed method is intuitive and sound. Moreover, it is easy to implement and does not cost excessive computation resources.
* The experiments show that the method is effective.

My concerns are as follows:
* The description of the method is not clear.
* The contribution of the paper is thin.

**Questions For The Authors:**

Where does the term $\hat{r}$ in Section 2.2 come from?

**Reasons To Accept:**

* The proposed method is intuitive and sound. Moreover, it is easy to implement and does not cost excessive computation resources.
* The experiments show that the method is effective.

**Reasons To Reject:**

* The description of the method is not clear. Specifically, the term $\hat{r}$ in Section 2.2 is not clearly explained. The source of this label is questionable.
* The contribution of the paper is thin. Essentially, this is a combination of the well-known re-ranking technique and the now popular reward models. Therefore, it does not provide interesting technical insights.

**Reproducibility:**

4: Could mostly reproduce the results, but there may be some variation because of sample variance or minor variations in their interpretation of the protocol or method.

**Reviewer Confidence:**

3: Pretty sure, but there's a chance I missed something. Although I have a good feel for this area in general, I did not carefully check the paper's details, e.g., the math, experimental design, or novelty.

---

> ### Author Rebuttal · Authors · 2023-08-27
>
> Hi, thank you for reviewing our paper! We decided to update the notations and add a pseudocode of RAD to enhance clarity.
>
>
> | Notation | Dimension | Description |
> |:-:|-|-|
> |$\beta$ | $\mathbb{R}^+$ | steering amount hyperparameter |
> |$l$ | $\mathbb{N}$ | generation length of reward model training data |
> |$\hat r$ | $[0,1]$ | label of reward model training data |
> |$\mathbf{r}$ | $[0,1]^l$ | predictions generated by reward model during training |
> |$\mathbf{w}_t$ | $\mathbb{N}^k$ | indices of top-$k$ tokens at time $t$ |
> |$\mathbf{z}_t$ | $\mathbb{R}^k$ | logits of top-$k$ tokens at time $t$ |
> |$\boldsymbol{\rho}_t$ | $[0,1]^k$ | reward scores predicted by the reward model at time $t$ |
>
> **Table 1:** Updated notations. We use two different notations $\mathbf{r}$ and $\boldsymbol{\rho}$ to differentiate the reward model's output in train time and test time.
>
> &nbsp;
>
> ---
> Algorithm 1:  Reward-Augmented Decoding
> ---
> **Input:** $f_\theta$, neural network language model (outputs logits) \
> &nbsp;&nbsp;&nbsp;&nbsp;&nbsp;&nbsp;&nbsp;&nbsp;&nbsp;&nbsp;&nbsp;&nbsp;&nbsp;$g_\lambda$, neural network reward model \
> &nbsp;&nbsp;&nbsp;&nbsp;&nbsp;&nbsp;&nbsp;&nbsp;&nbsp;&nbsp;&nbsp;&nbsp;&nbsp;$X$, generation prefix
>
> 1. $x_t \leftarrow \mathtt{none}$
> 2. **while** $x_t \ne \mathtt{<EOS>}$ **do**
> 3. &nbsp;&nbsp;&nbsp;&nbsp;
>     $\mathbf{w}_ t \leftarrow \texttt{topk}(f_\theta(X))$
>     &nbsp;&nbsp;&nbsp;&nbsp;&nbsp;&nbsp;&nbsp;&nbsp;&nbsp;&nbsp;&nbsp;&nbsp;
>     // get top-$k$ tokens (indices), $\mathbf{w}_t \in \mathbb{N}^k$
> 4. &nbsp;&nbsp;&nbsp;&nbsp;
>     $\mathbf{z}_ t \leftarrow f_\theta(X)[\mathbf{w}_t]$
>    &nbsp;&nbsp;&nbsp;&nbsp;&nbsp;&nbsp;&nbsp;&nbsp;&nbsp;&nbsp;&nbsp;&nbsp;&nbsp;&nbsp;&nbsp;&nbsp;&nbsp;&nbsp;&nbsp;
>     // get top-$k$ token logits, $\mathbf{z}_t \in \mathbb{R}^k$
> 5. &nbsp;&nbsp;&nbsp;&nbsp;
>     $\boldsymbol{\rho}_ t \leftarrow g_\lambda \left(\begin{bmatrix} X;\mathbf{w}_ {t,1} \\\\ \vdots \\\\ X;\mathbf{w}_ {t,k} \end{bmatrix}\right)$
>     &nbsp;&nbsp;&nbsp;&nbsp;
>     // compute rewards, $\boldsymbol{\rho}_t \in [0,1]^k$
> 6. &nbsp;&nbsp;&nbsp;&nbsp;
>     $p_t \leftarrow \mathtt{softmax}(\mathbf{z}_t+\beta \boldsymbol{\rho}_t)$
>     &nbsp;
>     // compute reweighted distribution
> 7. &nbsp;&nbsp;&nbsp;&nbsp;
>     $x_t \sim \mathrm{Categorical}(p_t)$
> 8. &nbsp;&nbsp;&nbsp;&nbsp;
>     $X \leftarrow \\{X;x_t\\}$
>     &nbsp;&nbsp;&nbsp;&nbsp;&nbsp;&nbsp;&nbsp;&nbsp;&nbsp;&nbsp;&nbsp;&nbsp;&nbsp;&nbsp;&nbsp;&nbsp;&nbsp;&nbsp;&nbsp;&nbsp;&nbsp;&nbsp;&nbsp;
>     // append new sample
>
> **Output:** generated text $X$ steered towards higher rewards
>
> ---
>
> &nbsp;
>
>
> Since you mentioned that the method description is not clear, we provide a summary of section 2.1 and 2.2 below:
> - (2.1 paragraph 1) We use a small, decoder-only Transformer model, with a linear layer on top, as a reward model to guide generation. The decoder-only model’s causal masking enables a lower complexity during decoding.
> - (2.1 paragraph 2) The reward model is trained on a dataset (which can differ from the target dataset) with a weighted, cumulative squared error loss. This loss encourages the reward model to converge to the ground-truth reward early in the text sequence.
> - (2.2) At each generation timestep $t$, we consider the top-$k$ tokens as candidates; record their logits $\mathbf{z}_t$; append candidate tokens to current text sequence to form $k$ candidate sequences; evaluate the rewards $\boldsymbol{\rho}_t$ of these candidate sequences; finally, reweight token probabilities with
> $\mathtt{softmax}(\mathbf{z}_t + \beta \boldsymbol{\rho}_t)$.
>
> To address your question about $\hat r$, it is the label from the dataset used to train the reward model, which usually represents how well a text sequence aligns with the desired attribute.
>
> Regarding your comment about not providing interesting technical insights, we agree that reward modeling and weighted decoding are two well-studied techniques—reward model has widely been used in reinforcement learning and methods that involve data exploration, and weighted decoding is an intuitive technique backed by a Bayesian interpretation. However, prior work has neglected the idea of connecting reward modeling and weighted decoding when it comes to aligning language generation, partly due to the prohibitive complexity incurred by a bi-directional, discriminative reward model. In this paper, we draw a straight relation between reward modeling and weighted decoding by using a uni-directional reward model to augment the decoding process without re-training the base language model. We show that RAD is simple, efficient, and easily generalizable. We therefore believe its merit is self-evident. We further highlight that connecting reward modeling and weighted decoding is a novel contribution, as is the innovation of using a causal reward model in this setting to reduce computational costs. Furthermore, RAD enables a new form of modularity where a given reward model can be immediately integrated into a language model to steer its generations. Coupled with the small size of the reward model, this could enable language model users to cheaply create custom-tailored reward models that could be applied to different language models (including those behind black-box APIs).

---

### Official Review · Reviewer_U1EG · 2023-08-05

**Soundness:** 4

**Excitement:**

4: Strong: This paper deepens the understanding of some phenomenon or lowers the barriers to an existing research direction.

**Paper Topic And Main Contributions:**

This paper studies efficient controlled text generation by a lightweight external module for large language models. At each time step of text generation, the proposed method unidirectionally classifies the attributes of candidate texts and modifies the logit of the language model according to the scores. Experimental results show that the proposed weighting module achieves better performance in controlled text generation than previous methods.

**Reasons To Accept:**

1. The proposed method is simple yet effective and thus should influence subsequent research in controlled text generation.
2. The proposed method achieves a good trade-off between efficiency and effectiveness compared to previous weighted decoding and fine-tuning methods.
3. The results of the experiment are reliable because of the extensive comparison with previous models and the recalculation of the scores of previous models based on nice attention to the differences in API.

**Reasons To Reject:**

I found no major reasons for rejection, but it would be more insightful to explain why the proposed method performs better than other weighted decoding methods.

**Reproducibility:**

4: Could mostly reproduce the results, but there may be some variation because of sample variance or minor variations in their interpretation of the protocol or method.

**Reviewer Confidence:**

3: Pretty sure, but there's a chance I missed something. Although I have a good feel for this area in general, I did not carefully check the paper's details, e.g., the math, experimental design, or novelty.

---

> ### Author Rebuttal · Authors · 2023-08-27
>
> Thank you for your review! Below, we provide some insights into why RAD performs better compared to other weighted decoding methods.
>
> First, note the different uses of the small auxiliary model in RAD compared to other strong weighted decoding methods such as GeDi and DExperts—we use the auxiliary model as a reward model to discriminate candidate sequences at each intermediate step whereas GeDi and DExperts use the auxiliary model as a class-conditional language model to steer generation. One reason that RAD's approach works better could be that classification is an easier task as compared to class-conditional language modeling, allowing small discriminators to be more helpful than small expert LMs in guiding language generation.
>
> Second, RAD’s reward model makes more use of label information if it is not a binary score (i.e. a fraction in [0,1]). RAD’s reward model is trained to minimize the squared error loss while the auxiliary model in GeDi and DExperts does not utilize the label score efficiently—they use label score primarily for data filtering for generative training.
>
> Another factor that may have contributed to RAD’s superior performance is the weighted cumulative loss mentioned in section 2.1. During reward model training, by taking every prefix’s reward score into account, we encourage the reward model to converge to the correct reward score early in the sequence. In this way, the reward model is trained not only to capture the current but also the future alignment score of a generation course, which better steers the generation.
>
> We believe RAD is exciting in that it is simple, efficient, and easily generalizable. We further highlight that connecting reward modeling and weighted decoding is a novel contribution, as is the innovation of using a causal reward model in this setting to reduce computational costs. Moreover, RAD enables a new form of modularity where a given reward model can be immediately integrated into a language model to steer its generations. Coupled with the small size of the reward model, this could enable language model users to cheaply create custom-tailored reward models that could be applied to different language models (including those behind black-box APIs).

---

### Meta-Review · Area_Chair_rdNA · 2023-09-20

**Recommendation:** 4

**Metareview:**

The paper presents a method to more effectively do controlled generation by using a reward model as a heuristic during a search-based decoding process. The method is a natural extension of multiple existing approaches and the reviews all agree for the most part that this work is both exciting and has sound methodology.

---

### Decision · Program_Chairs · 2023-10-07

**Decision:**

Accept-Main

**Comment:**

The paper presents a method to more effectively do controlled generation by using a reward model as a heuristic during a search-based decoding process. The method is a natural extension of multiple existing approaches and the reviews all agree for the most part that this work is both exciting and has sound methodology.